# Non-autoregressive Text Editing with Copy-aware Latent Alignments

**Yu Zhang**[☆*]  **Yue Zhang**[☆*]  **Leyang Cui**[☾]  **Guohong Fu**[☆†]

[☆]Institute of Artificial Intelligence, School of Computer Science and Technology,
Soochow University, Suzhou, China
[☾]Tencent AI Lab
yzhang.cs@outlook.com; yzhang21@stu.suda.edu.cn
leyangcui@tencent.com; ghfu@suda.edu.cn
⌂ https://github.com/yzhangcs/ctc-copy

## Abstract

Recent work has witnessed a paradigm shift from Seq2Seq to Seq2Edit in the field of text editing, with the aim of addressing the slow autoregressive inference problem posed by the former. Despite promising results, Seq2Edit approaches still face several challenges such as inflexibility in generation and difficulty in generalizing to other languages. In this work, we propose a novel *non-autoregressive* text editing method to circumvent the above issues, by modeling the edit process with latent CTC alignments. We make a crucial extension to CTC by introducing the copy operation into the edit space, thus enabling more efficient management of textual overlap in editing. We conduct extensive experiments on GEC and sentence fusion tasks, showing that our proposed method significantly outperforms existing Seq2Edit models and achieves similar or even better results than Seq2Seq with over $4\times$ speedup. Moreover, it demonstrates good generalizability on German and Russian. In-depth analyses reveal the strengths of our method in terms of the robustness under various scenarios and generating fluent and flexible outputs.

## 1 Introduction

In natural language processing, *monolingual* text generation involves producing a target sequence from the source text with significant textual overlap (Malmi et al., 2022). This includes a range of text-editing tasks such as grammatical error correction (GEC) (Ng et al., 2014) and sentence fusion (Geva et al., 2019), as shown in Table 1.

Generally, text editing can be addressed under the standard *Seq2Seq* framework (Lebanoff et al., 2020; Rothe et al., 2021). Despite their decent performance, Seq2Seq has been criticized (Sun et al., 2021) for its inferior inference speed due to the

---

*Work was done during the internship at Tencent AI Lab. Yu Zhang and Yue Zhang make equal contributions.

†Corresponding author.

Grammatical Error Correction
| | |
|---|---|
| *source:* | ~~Me~~ want to go store. |
| *target:* | *I* want to go *to the* store. |

Sentence Fusion
| | |
|---|---|
| *source:* | ~~The~~ sun set~~. The~~ sky darkened. |
| *target:* | *As the* sun set, *the* sky darkened. |

Table 1: Text editing examples for grammatical error correction and sentence fusion.

autoregressive generation fashion, i.e., generating tokens one by one. Consequently, the practical applications of Seq2Seq models are limited in modern online text assistance systems.

To overcome the above deficiency, recently, there is a growing interest in an alternative approach, referred to as *Seq2Edit* (Awasthi et al., 2019; Omelianchuk et al., 2020; Mallinson et al., 2020), which, in contrast, proposes to reconstruct the target sentence by applying a set of edit operations, e.g., keep, deletion and insertion, to the input. Drawing on the insight that the input/output tokens are heavily shared, Seq2Edit favors copying most of the source text directly via the keep operation, which eases the reliance for an autoregressive decoder (Malmi et al., 2019; Mallinson et al., 2022). Among others, the best-performing GECToR (Omelianchuk et al., 2020, 2021) directly formulates text-editing as a non-autoregressive sequence tagging task, thus enabling more efficient parallelizable inference. GECToR demonstrates remarkable results on many tasks, meanwhile being orders of magnitude faster than its autoregressive counterparts (Rothe et al., 2020).

However, several challenges arise as we try to have the cake and eat it. We argue that Seq2Edit works represented by GECToR still suffer from two main issues:

i **Flexibility**: Seq2Edit learns to edit text by predefining a fixed and relatively small (e.g., 5,000) edit vocabulary collected from the training data, which is at the sacrifice of generation flexibility.

ii **Language generalization**: Seq2Edit needs to delve into linguistic features to customize the edit actions, e.g., VB-VBZ for subject-agreement edits and PLURAL for singular-plural form conversions, thus diminishing its ability to generalize to other languages.

Our desiderata in this work is to design a *non-autoregressive* model for text editing that enjoys the merits of both efficiency and effectiveness, meanwhile generalizing well to other languages. This poses two considerations: 1) flexible, non-manually defined edit space; 2) a minimal set of tailored operations (Dong et al., 2019) to maintain the generalization. Taking inspirations from recent progresses in non-autoregressive text generation (Libovický and Helcl, 2018; Saharia et al., 2020; Huang et al., 2022b), in this work, we propose a novel method for text editing that meets the aforementioned expectations by making a direct yet effective extension to connectionist temporal classification (CTC) (Graves et al., 2006).

Unlike previous works focusing on generating arbitrary tokens (Gu and Kong, 2021), the key insight here is to interpret the vanilla CTC alignment as an executable edit sequence, primarily composed of two kinds of operations: DELETE and ADD$_t$. This perspective opens the door for combining the alignment with the edit actions in existing Seq2Edit works. Specifically, we further extend the alignment space by incorporating KEEP, a label used to facilitate direct copy of the respective source tokens. We find it is essential for processing textual overlap in editing, yielding significant performance gains. During training, our method marginalizes out all (valid) latent edit alignments to maximize the likelihood of the target text (Graves et al., 2006). During inference, like GECToR, it simply takes the token with highest probability as the output for each position simultaneously (see Table 2), ensuring the high efficiency. The contributions of this work are four-fold:

- We propose a novel method, that extends CTC with the copy operation to address edit-based text generation. To the best of our knowledge, this is the first attempt to adapt CTC to deal with text editing tasks.
- We conduct experiments on GEC and sentence fusion, and find that our proposed model performs favorably better than all existing Seq2Edit models, meanwhile showcasing good generalization capabilities in multilingual settings.

- We show that our model achieves similar or even better results compared to Seq2Seq across all experiments with $\sim 4\times$ speedup.
- Extensive analyses on our method reveal its merits in terms of robustness under different scenarios as well as the superiority in generation flexibility against existing systems.

## 2 Preliminaries

We begin by introducing some notations. The goal for text-editing is to transform the source sentence $\boldsymbol{x} = x_0, x_1, \ldots, x_N$ into the desired target $\boldsymbol{y} = y_0, y_1, \ldots, y_M$ with $N$ and $M$ tokens, respectively.

**Connectionist Temporal Classification** was first introduced in auto speech recognition (ASR) (Graves et al., 2006), aiming to circumvent the problems of no explicit alignments between ASR inputs/outputs. Specifically, CTC introduces a special blank token $\varnothing$ on top of the vocabulary $\mathcal{V}$, and defines a latent alignment path $\boldsymbol{a} = a_0, a_1, \ldots, a_N$ between $\boldsymbol{x}$ and $\boldsymbol{y}$ with $a_i \in \mathcal{V} \bigcup \{\varnothing\}$, which is of equal length as $\boldsymbol{x}$. During training, CTC models the probability of the target sequence by marginalizing the probabilities of all latent alignments,

$$P(\boldsymbol{y} \mid \boldsymbol{x}) = \sum\nolimits_{\boldsymbol{a} \in \Gamma(\boldsymbol{y})} P(\boldsymbol{a} \mid \boldsymbol{x}) \qquad (1)$$

where $\Gamma(\cdot)$ is a mapping function that returns all possible alignment paths. CTC views each $a_i \in \boldsymbol{a}$ independent of each other and factorizes the probability of $\boldsymbol{a}$ as

$$P(\boldsymbol{a} \mid \boldsymbol{x}) = \prod\nolimits_{a_i \in \boldsymbol{a}} P(a_i \mid \boldsymbol{x}) \qquad (2)$$

In this way, CTC permits very efficient calculation of Eq. 1 in $\mathcal{O}(N \times M)$ via forward algorithm. We refer interested readers to the original paper (Graves et al., 2006) and tutorials by Hannun (2017) for more details.

During inference, CTC defines a collapsing function $\Gamma^{-1}(\cdot)$ to recover the target sequence $\boldsymbol{y}$ from $\boldsymbol{a}$ by removing all blanks and *consecutive* repetitive tokens. For example, assuming a possible alignment path $\boldsymbol{a} = \{\mathrm{a}, \mathrm{a}, \varnothing, \mathrm{a}, \mathrm{b}, \mathrm{b}\}$, then $\Gamma^{-1}(\boldsymbol{a})$ returns $\{\mathrm{a}, \mathrm{a}, \mathrm{b}\}$.

**Non-autoregressive text generation** (NAT) differs from its autoregressive counterpart in that it generates all target tokens simultaneously rather than one-by-one. NAT often runs several times faster than autoregressive Seq2Seq models as it is highly parallelized. Very recently, CTC has been

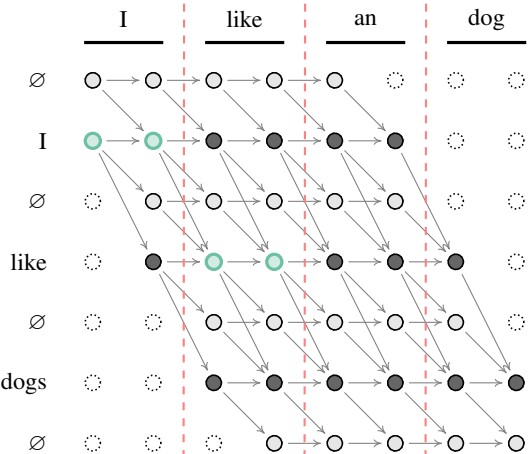

Figure 1: A running GEC example by CTC with an upsampling ratio of 2, which corrects the phase "*an dog*" in the source sentence to "*dogs*". Each source token is separated by red-dashed lines. Grey and black nodes represent blanks ($\varnothing$) and normal tokens respectively. Green nodes indicate the positions where the source tokens can be copied directly. The arrows represent all valid transition paths.

| $\boldsymbol{x}$ | I | | like | | an | | dog | |
|---|---|---|---|---|---|---|---|---|
| $\boldsymbol{a}$ | K | K | K | K | $\varnothing$ | $\varnothing$ | $\varnothing$ | dogs |
| | I | ~~I~~ | like | ~~like~~ | $\varnothing$ | $\varnothing$ | $\varnothing$ | dogs |
| $\boldsymbol{y}$ | I | | like | | | | | dogs |

Table 2: An inference example for editing the source sentence $\boldsymbol{x}$. The output $\boldsymbol{a}$ is produced by our copy-aware CTC, which predicts an 1-best token for each position with the green label K denoting the copy label. The output $\boldsymbol{y}$ is the final recovered result by the collapsing function $\Gamma^{-1}(\cdot)$.

where each $\boldsymbol{r}_i \in \mathbb{R}^H$, $H$ is the size of the hidden vector. Once the hidden states are obtained, we employ a simple linear projection followed by two Transformer decoder layers to upsample each $\boldsymbol{h}_i$ to $T$ new sample vectors, ensuring that the scaled input, which is $T\times$ as long as the the source, is strictly longer than the desired output

$$\boldsymbol{h}_{iT+1}, \ldots, \boldsymbol{h}_{iT+T} = \texttt{Decoder}(W\boldsymbol{r}_i + b) \quad (4)$$

where $W \in \mathbb{R}^{TH \times H}, b \in \mathbb{R}^{TH}$ are learnable parameters. We fix the value of upsampling ratio $T$ to 4 in this work after careful ablation analyses (§ 5.1). In this way, we can generate target sentences by CTC with very flexible length control. We employ another linear layer followed by the softmax function over $\boldsymbol{h}_i$ to obtain $P(a_i \mid \boldsymbol{x})$.

## 3.2 Copy-aware CTC

The output space of vanilla CTC comprises the general vocabulary $\mathcal{V}$ as well as the blank token $\varnothing$. We can utilize CTC to mimic the edit processes by symbolizing generating a token $\mathtt{t} \in \mathcal{V}$ as $\texttt{ADD}_t$, representing the insertion operation, and $\varnothing$ as $\texttt{DELETE}$, meaning deleting a source token. This satisfies the aforementioned desiderata of learning to edit with a minimal set of operations (Dong et al., 2019), and maintaining enough flexibility by means of marginalizing all latent alignments defined over the entire vocabulary. However, vanilla CTC is still wasteful for text editing as it lacks explicit modeling of the copy behavior. We in this work propose to bridge this gap by introducing a special token K to denote the KEEP operation. Concretely, we interpret generating K at $a_{iT+j}$, the $j$th upsampled position for $i$th source token, as directly copying the source token $x_i$. In this way, the final output space of each $a_i$ is $\mathcal{V} \bigcup \{\mathtt{K}\} \bigcup \{\varnothing\}$.

**Training objective** Our final objective is to minimize the negative log-likelihood of all possible

---

introduced to non-autoregressive NMT (Libovický and Helcl, 2018; Saharia et al., 2020; Gu and Kong, 2021). In ASR, CTC assumes the input length $N$ is larger that the output length $M$ so that we can safely delete blanks from the alignment, resulting in a shorter output sequence. However, this is not the fact in text generation. To remedy this, Libovický and Helcl (2018) propose to make use of an upsampling layer to amplify the input first and then run CTC as usual. This enables the model to learn target lengths very flexibly, which we believe is an important factor that empowers the CTC model.

## 3 Methodology

In this section, we will introduce our proposed method that adapts the vanilla CTC to text editing tasks. The main idea is to endow CTC with the ability of modeling the edit processes by extending the latent CTC alignments with interpretable edit operations, especially the copy operation.

### 3.1 Model

The basic architecture of our model is encoder-only. Given the input $\boldsymbol{x} = x_1, x_2, \ldots, x_N$, we simply take a pretrained language model (PLM) (Devlin et al., 2019) as the backbone encoder to obtain the contextualized representations.

$$\boldsymbol{r}_1, \boldsymbol{r}_2, \ldots, \boldsymbol{r}_N = \texttt{PLM}(x_1, x_2, \ldots, x_N) \quad (3)$$

alignments with the three kinds of edit operations

$$\mathcal{L} = -\log \sum_{\boldsymbol{a} \in \Gamma'(\boldsymbol{y})} P(\boldsymbol{a} \mid \boldsymbol{x}) \qquad (5)$$

where $\Gamma'(\cdot)$ is the new mapping function extending $\Gamma(\cdot)$ to KEEP. Assuming an input $\{a, a, b\}$ with $T = 2$, two possible paths returned by $\Gamma'(\cdot)$ can be $\{a, a, \varnothing, a, b, b\}$ and $\{K, K, \varnothing, K, K, K\}$.

**Glancing training**  Previous works have shown that the glancing training strategy (Qian et al., 2021) can give a boost to the performance of non-autoregressive generation. So we also adopt this method in our training process. The key idea is to sample some ground-truth tokens as inputs to the decoder to guide the model once the references are too difficult to fit, which is reminiscent of curriculum learning. Specifically, the objective becomes

$$\mathcal{L}_{\text{glancing}} = -\log P(\boldsymbol{y} \mid \boldsymbol{x}, \bar{\boldsymbol{h}})$$

where $\bar{\boldsymbol{h}}$ is obtained by replacing some hidden states $\boldsymbol{h}_i$ with the embeddings of sampled gold tokens. The sampling ratio is determined following three steps (Qian et al., 2021): 1) determine a gold alignment compatible with the target by Viterbi decoding (Chan et al., 2020; Huang et al., 2022b; Shao et al., 2022): $\boldsymbol{a} = \arg\max_{\boldsymbol{a} \in \Gamma'(\boldsymbol{y})} P(\boldsymbol{a} \mid \boldsymbol{x})$; 2) decode an 1-best alignment path $\boldsymbol{a}'$ ; 3) then the ratio becomes $p = \tau \sum_i [\boldsymbol{a}_i = \boldsymbol{a}'_i]$, which is proportional to the number of different tokens between the predicted sequence and the gold alignment. We set $\tau$ to 1 in the following experiments.

### 3.3  Inference

During inference, we directly predict the 1-best token for each position in parallel, followed by the post-processing process. Taking Table 2 as an example, first, an alignment path $\boldsymbol{a}' = \{K, K, K, K, \varnothing, \varnothing, \varnothing, \text{dogs}\}$ is produced by finding the 1-best token for each position greedily. Then each label K is translated to the corresponding source token. Finally, we can successfully recover the output with the collapsing function, i.e., $\Gamma'^{-1}(\boldsymbol{a}') = \Gamma^{-1}(\text{I, I, like, like}, \varnothing, \varnothing, \varnothing, \text{dogs}\}) = \{\text{I, like, dogs}\}$.

**Iterative decoding**  Following Omelianchuk et al. (2020), we also employ the techniques of iterative decoding to better capture the edits hard to make in one pass. We simply take the collapsed output of CTC as the model input during the next iteration (Awasthi et al., 2019). In pratice, we found that it

brought considerable performance gains, but the improvements saturate gradually after 2 iterations. So we choose to uniformly refine the outputs twice for a good speed-performance tradeoff.

## 4  Experiments

Following FELIX and EDIT5 (Mallinson et al., 2020, 2022), we evaluate our model by conducting experiments on two text editing tasks: grammatical error correction (GEC) and sentence fusion, both of which are representative and have sufficient data for training. We plan to conduct examinations on more tasks in future work due to space limitations.

### 4.1  Grammatical Error Correction

The task of grammatical error correction involves detecting and correcting the grammatical errors in a given sentence.

**Setup**  For English, we adopt a 3-stage training strategy to train our GEC models (Zhang et al., 2022a): 1) pretrain the model on CLANG-8 (Rothe et al., 2021), a cleaned version of the LANG-8 data; 2) finetune the pretrained model on the combination of three datasets, namely FCE (Yannakoudakis et al., 2011), NUCLE (Dahlmeier et al., 2013) and W&I+LOCNESS (Bryant et al., 2019); 3) finally, we further finetune the model on the high-quality W&I+LOCNESS. During training, we use BEA19 Dev data as the validation set. We evaluate our models by reporting P/R/$F_{0.5}$ points on BEA19 Dev data using the ERRANT toolkit (Bryant et al., 2017) and CoNLL14 Test data (Ng et al., 2014) using M2Scorer (Dahlmeier and Ng, 2012). Besides, without additional training, we also report GLEU scores on JFLEG Test data (Napoles et al., 2017) to measure the fluency of CTC-generated texts. More details on data statistics and training details are available in § A.

**Results**  We present the main GEC results in Table 3. It is hard to make fully fair comparisons with existing works as the training data varies vastly, which has a huge impact on the final results (Omelianchuk et al., 2020). We therefore re-implemented BART-based Seq2Seq and GEC-ToR and ran them under the same environments for more comparable results. In the top group of the Table, first, we observe that our re-implemented BART achieves 68.2 $F_{0.5}$ score on CoNLL14, outperforming the cutting-edge SynGEC (Zhang et al., 2022b); second, our CTC is superior to BART

| | PLMs | BEA19 | | | CoNLL14 | | | JFLEG | Speedup |
|---|---|---|---|---|---|---|---|---|---|
| | | P | R | $F_{0.5}$ | P | R | $F_{0.5}$ | | |
| *Autoregressive* | | | | | | | | | |
| Kaneko et al.♣ | - | - | - | - | 69.3 | 45.0 | 62.6 | 61.3 | - |
| SAD (Sun et al.)♣ | BART | - | - | - | 71.0 | 52.8 | 66.4 | - | 3.8× |
| Seq2Seq (Zhang et al.) | BART | 63.1 | 44.8 | 58.3 | 73.6 | 48.6 | 66.7 | 61.5 | - |
| SynGEC (Zhang et al.)♠ | BART | 64.5 | **45.7** | **59.6** | 74.7 | 49.0 | 67.6 | 62.2 | - |
| ChatGPT (Fang et al.) | - | - | - | - | 51.3 | **62.4** | 63.2 | **63.5** | - |
| Seq2Seq† | BART | 65.0 | 40.7 | 58.0 | **76.0** | 48.4 | **68.2** | 60.2 | 1.0× |
| *Non-autoregressive* | | | | | | | | | |
| LaserTagger (Malmi et al.) | BERT | - | - | - | 50.9 | 26.9 | 43.2 | - | - |
| GECToR (Omelianchuk et al.)♣ | RoBERTa | - | - | - | 75.3 | 44.4 | 66.1 | - | - |
| GECToR† | RoBERTa | **66.2** | 36.5 | 56.9 | 73.2 | 51.1 | 67.4 | 57.6 | 2.9× |
| Ours | RoBERTa | 62.2 | **46.9** | 58.4 | 74.9 | 50.6 | 68.3 | 62.4 | **4.1×** |

Table 3: Main results on BEA19 Dev, CoNLL14 Test, and JFLEG Test data. Our results are averaged over 4 runs with different random seeds. ♣ means using external or synthetic data for pretraining; ♠ uses external syntactic knowledge, and is thus incomparable. † means the results are obtained from running our re-implemented code.

on all datasets by 0.4, 0.1 and 2.2, respectively. On BEA19 Dev data, CTC surpasses all previous works except SynGEC, which is enhanced by external syntax trees. On JFLEG Test data, our model exhibits a GLEU score 62.4, second only to ChatGPT (Fang et al., 2023), which is very close to human-like performance, indicating that our model excels at generating fluent sentences. In the bottom group, we can see that our CTC greatly surpasses the best-performing GECToR by 1.5, 0.9 and 4.8 on BEA19 Dev, CoNLL14 Test and JFLEG Test data, respectively, achieving new state-of-the art in the area of non-autoregressive text-editing.

**Speed Comparisons** We compare different models in terms of inference speed on CoNLL14 in the last column of Table 3. For fair comparisons, all of our models are run on a single Nvidia Tesla V100 GPU with roughly 10,000 tokens per batch. We use BART-based Seq2Seq with decoding beam size of 12 as the speed benchmark. We incorporate the KV cache trick (Pope et al., 2023) to eliminate redundant computations.[1] It takes about 45 seconds to parse all 1,312 CoNLL14 sentences. As we can see, our CTC delivers a 4.1× speedup against Seq2Seq and is even faster than GECToR (2.9×), which also operates non-autoregressively. It owes much to the fact that CTC requires fewer iterations of refinement. SAD (Sun et al., 2021) achieves similar efficiency to ours but with a much smaller (12+2) model size. Overall, we can conclude that our model performs orders of magnitude faster than Seq2Seq under similar conditions, readily meeting

| | EM | SARI |
|---|---|---|
| *Autoregressive* | | |
| Transformer (Geva et al.) | 51.1 | 84.5 |
| LaserTagger$_{AR}$ (Malmi et al.) | 53.8 | 85.5 |
| Seq2Edits (Stahlberg and Kumar) | 61.71 | 88.73 |
| BERT$_{share}$ (Rothe et al.) | 65.3 | 89.9 |
| RoBERTa$_{share}$ (Rothe et al.) | **66.6** | **90.3** |
| *Non-autoregressive* | | |
| LaserTagger$_{FF}$ (Malmi et al.) | 52.2 | 84.1 |
| FELIX (Mallinson et al.) | 61.31 | 88.78 |
| EDIT5 (Mallinson et al.) | 64.95 | - |
| Ours | **66.0** | **90.7** |

Table 4: Sentence fusion results on DiscoFuse Test data. BERT$_{share}$ and RoBERTa$_{share}$ are Seq2Seq models but initialized with 24-layer BERT and RoBERTa weights, respectively.

the demands of online inference.

## 4.2 Sentence Fusion

Sentence fusion is the task of fusing several independent sentences into a single coherent text.

**Setup** We train our sentence fusion models on the balanced Wikipedia portion of DiscoFuse data (Geva et al., 2019) following Mallinson et al. (2020, 2022). For evaluation, we report two metrics (Geva et al., 2019), i.e., **E**xact **M**atch (EM), which measures the percentage of exactly correct predictions, and SARI (Xu et al., 2016), which computes the averaged F1 scores of the inserted, kept, and deleted n-grams. For consistency, we use Geva et al. (2019)'s implementation[2] to compute SARI.

**Results** are listed in Table 4. We can see that our model surpasses all non-autoregressive works significantly, especially EDIT5, by more than 1 point

---

[1] We observe that the KV cache trick brings a significant speedup. Our CTC model performs over 30× faster than the basic implementation without the trick.

[2] https://git.io/fj8Av

| | $T$ | BEA19 | | |
|---|---|---|---|---|
| | | P | R | $F_{0.5}$ |
| CTC (vanilla) | 4 | 57.9 | **46.1** | 55.1 |
| CTC (ours) | 4 | **61.8** | 43.4 | **57.0** |
| - GLAT | 4 | 60.2 | 44.2 | 56.1 |
| | 2 | 60.9 | 43.8 | 56.5 |
| | 6 | 60.9 | 44.2 | 56.6 |
| | 8 | 60.7 | 45.1 | 56.8 |

Table 5: The results of vanilla CTC and our proposed variant on BEA19 Dev at stage 1. "$T$" stands for the upsampling ratio. "- GLAT" means removing the GLAT trick during training.

EM score. One key observation is that 10.5% out of 4.5M fusion examples require source reordering. LaserTagger (Malmi et al., 2019) deals with this by defining a SWAP operation while EDIT5 uses pointer networks instead. The results indicate that our model is capable of doing reordering implicitly and thus handles the fusion task skillfully. On the other hand, our model achieves final EM/SARI scores of 66.0/90.7, showing strong competitiveness with the best performing RoBERTa$_{\text{share}}$ (66.6/90.3).

## 5 Analysis

We demonstrate the superiority of our proposed CTC model by making comparisons from two perspectives: 1) with vanilla CTC; 2) with other text-editing systems.

### 5.1 Comparisons with Vanilla CTC

**Ablation** We study the effectiveness of our proposed copy-aware CTC in Table 5. It is clear that our model brings remarkable gains over the vanilla CTC, especially in terms of precision, by 4 points. This suggests that introducing the copy operation can effectively suppress the over-confident revisions of vanilla CTC, thus greatly reducing the errors. We also study the impact of the GLAT trick and upsampling ratios in the Table. We can conclude that GLAT has a non-negligible contribution (0.9) to the final results. Additionally, as the upsampling ratio $T$ grows from 2 to 8, the results increase from 56.1 to 57.0 and later diminish; the optimal ratio was found to be 4.

**Convergence behavior** In Fig. 2, we plot the training curves regarding $F_{0.5}$ scores and iterations of the two models. From the figure, we clearly see that it took about 10 iterations for our copy-aware CTC to reach the peak results, while 40 iterations for vanilla CTC. Our proposed CTC variant converges much faster than the vanilla one, with a final

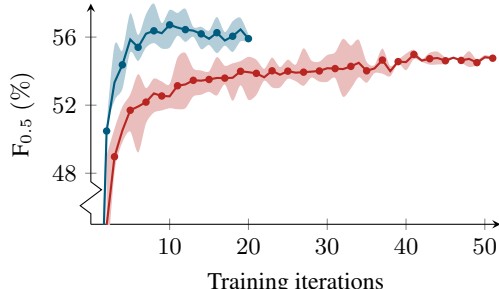

Figure 2: The training curves of vanilla CTC (red) and our CTC variant (blue), respectively. We plot the averaged $F_{0.5}$ scores at stage 1 on BEA19 Dev data at each iteration, along with the upper/lower bound for different runs.

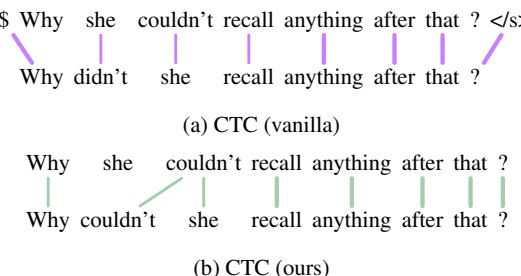

(a) CTC (vanilla)

(b) CTC (ours)

Figure 3: Two predictions made by vanilla CTC and CTC (ours), respectively. The connected lines are produced alignments. Green lines signify tokens directly copied from the source. The line thickness is decided by the confidence of the prediction. The correct edit is "*she couldn't → couldn't she*", while vanilla CTC wrongly corrects "*couldn't*" to "*didn't*".

gain of 1.5 $F_{0.5}$ points.

**Alighnment behavior** We give some prediction examples made by vanilla CTC and our proposed CTC variant in Fig. 3. We draw two observations from the figure: 1) in contrast to vanilla CTC, our proposed CTC variant copies most of the tokens in the prediction from the source text, thereby reducing the *over-correction* phenomenon to some extent and respecting the *minimum edit* principle of GEC (Ng et al., 2014); 2) the predicted alignments of our proposed CTC variant are more in agreement with human opinions than those of vanilla CTC. We attribute the difference largely to the copy operation, which serves as a pivot to guide the model on how to align with the source tokens, thereby allowing for more sensible edits.

### 5.2 Comparisons across Different Systems

We conduct a series of comparisons here between our proposed CTC, GECToR (a representative Seq2Edit model), and BART-based Seq2Seq, to gain a deeper understanding of the pros and cons of our model.

| | German | | | Russian | | |
|---|---|---|---|---|---|---|
| | P | R | $F_{0.5}$ | P | R | $F_{0.5}$ |
| *Autoregressive* | | | | | | |
| Náplava et al.♣ | 78.21 | 59.94 | 73.71 | 63.26 | 27.50 | 50.20 |
| Sun et al.♣ | 74.31 | 61.46 | 71.33 | 61.40 | 27.47 | 49.24 |
| gT5$_{xxl}$♣ | - | - | 75.96 | - | - | 51.62 |
| mBART♣ | - | - | - | 53.50 | 26.35 | 44.36 |
| mBART | 73.97 | 53.98 | 68.86 | 32.13 | 4.99 | 15.38 |
| mT5$_{base}$ | - | - | 67.19 | - | - | 25.20 |
| mT5$_{large}$ | - | - | 70.14 | - | - | 27.55 |
| *Non-autoregressive* | | | | | | |
| CTC | 71.2 | 57.8 | 68.0 | 36.0 | 14.0 | 27.3 |

Table 6: Multilingual GEC results on German Falko-MERLIN Test data and Russian RULEC-GEC Test data. Our model is trained on 24-layer XLM-RoBERTa, which is similar in scale to mBART & mT5$_{base}$ (12+12), and is smaller than mT5$_{large}$ (24+24). ♣ means using synthetic data for pretraining. mBART: Katsumata and Komachi (2020); gT5, mT5: Rothe et al. (2021).

**Multilingual results** To validate if CTC can be well generalized to other languages, we conduct multilingual GEC experiments on German and Russian, using their own portions of CLANG-8 data for training and Falko-MERLIN & RULEC-GEC Test data for evaluation, respectively. The results are presented in Table 6, where GECToR results are absent as we are aware of no GECToR extensions for these languages until now.[3] We can see that our CTC performs similar to (m)BART on German and surpasses it by 9 $F_{0.5}$ points on Russian. Furthermore, CTC outperforms similar sized mT5$_{base}$ by 0.1 and 2.1 $F_{0.5}$ points on German and Russian, and is on par with mT5$_{large}$ on Russian data, demonstrating the effectiveness of our method in multilingual settings. We highlight that, to the best of our knowledge, we are the first *non-autoregressive* model that performs on par with Seq2Seq counterparts on multilingual data. It is also worth noting that our models are purely trained on CLANG-8 and we do not pursue any data-augmentation tricks for further enhancements as it is beyond the scope of this work. We believe that our model can be greatly improved by introducing more synthetic data or increasing the model size, which we leave for future work.

**Regarding WERs** We report $F_{0.5}$ scores of the three systems broken down by word error rate (WER) in Figure 4. WERs are computed by counting the number of substitutions, deletions and in-

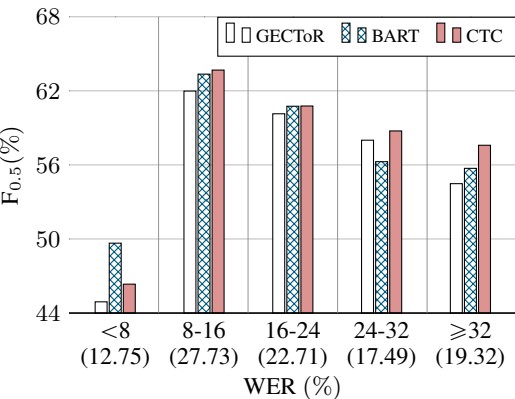

Figure 4: $F_{0.5}$ scores broken down by word error rate (WER) on BEA19 Dev data. Numbers in parentheses represent the percentage of gold edits in each group.

sertions required to edit the source sentence to the target, and dividing it by the number of tokens in the source. From the figure we can observe that the performance of BART is superior to CTC on sentences with low WERs, but tends to degrade as WERs increase. BART performs worse than CTC by a large margin when WERs⩾0.24, indicating that BART-based Seq2Seq models tend to make relatively conservative edits. This fact is also implied in Table 3, where BART exhibits higher precisions and lower recall scores, resulting an inferior overall score than CTC. On the other hand, there is a large gap between GECToR and CTC for WERs⩾0.32, showing that GECToR makes too stringent modifications and can hardly surpass CTC without several rounds of refinements in such scenarios.

**Regarding flexibility** It is known that the output flexibility of existing Seq2Edit models like GECToR is affected by the size of its small predefined vocabulary (Mallinson et al., 2022; Mesham et al., 2023), but to what extent? We conduct some quantitative analyses in Table 7 to answer this question. Specifically, we measure how well the three systems can do when producing edits out-of-GECToR vocabulary (OOV) by reporting their (token-level) ERRANT $F_{0.5}$ results. The scores are categorized into three primary coarse-grained error types, which correspond to the real edit operations in the GECToR vocabulary:

- ORTH & NOUN represents back-and-forth conversions of lower/upper cases and singular/plural forms, e.g., [*it*→ *It*] and [*citizen*→ *citizens*]. The associated GECToR operations involve only fixed string conversion rules and thus are not plagued by OOV problems.

---

[3]There are some pilot efforts to adapt GECToR to other languages, but with few exceptions (Zhang et al., 2022a), GECToR still hasn't achieved comparable performance to Seq2Seq yet. See discussions in their code issues.

| | ORTH & | VERB | | OTHER | |
|---|---|---|---|---|---|
| | NOUN | OOV (%) | Total | OOV (%) | Total |
| *Gold* | - | - (3.7) | - | - (8.0) | - |
| GECToR | **68.4** | - (0.0) | **59.0** | 19.3 (2.2) | 56.1 |
| BART | 67.9 | **27.1** (1.6) | 56.3 | **42.0** (4.2) | 58.5 |
| CTC | 67.9 | 24.8 (2.0) | 58.3 | 39.1 (5.5) | **59.5** |

Table 7: $F_{0.5}$ scores of the three systems on BEA19 Dev divided by different edit types. The column "OOV (%)" contains $F_{0.5}$ results of the corrections not covered by operations in the GECToR vocabulary and their corresponding percentages. The column "Total" contains overall $F_{0.5}$ scores for the type.

- VERB refers to verb form transformations like VB → VBZ ([*make→makes*]) and VB → VBD ([*draw→ drew*]), which are token-specific. GECToR pre-defines a very large verb-form vocabulary to handle these cases.
- OTHER refers to other miscellaneous types that fall into the basic $ADD_t$ or $REPLACE_t$ operations in GECToR, e.g., $REPLACE_a$ ([*The→a*]).

From Table 7, we observe that it's hard for GECToR to deal with the corrections not covered by the vocabulary.[4] Accordingly, the performance of GECToR is highly correlated with the proportion of OOVs in gold references. For the first two types, GECToR provides very good coverage, with only 3.7% of gold VERB cases not included in the operation space. GECToR performs favorably better than BART and CTC in this scenario. However, for OTHER, there is a considerable portion (8.0%) of gold references not contained in the vocabulary, and hence GECToR is greatly influenced. In contrast, both BART and our CTC can produce more flexible corrections, with around 4.2% and 5.5% edits that can not be produced from the limited GECToR operation space, leading to great improvements against GECToR.

## 6 Related Works

**Efficient text editing** For years, Seq2Seq models have been hitherto the best approaches for text editing (Vaswani et al., 2017; Lewis et al., 2020) due to their effectiveness. Some recent works obtain notable gains by injecting edit information into the generation process, making it more controllable and interpretable (Li et al., 2022). SynGEC (Zhang et al., 2022b) integrates edit templates into syntax trees, which are then encoded with GNNs in or-

---

[4]Strictly speaking, although difficult, it is possible for GECToR to make OOV corrections by multiple edit operations. For example, one can mimic the OOV $REPLACE_{Apples}$ by $UPPER_{apple}$ followed by PLURAL. That explains why the OOV $F_{0.5}$ scores for GECToR are not necessarily 0.

der to provide hints for the decoder. There is also a broad strand of works that combine the autoregressive fashion with edit operations (Stahlberg and Kumar, 2020; Reid and Zhong, 2021; Reid and Neubig, 2022, *inter alia*). Despite promising results, the slow inference speed limits their applications in real-life online systems. To combat this, Sun et al. (2021) make use of very shallow decoders as well as aggressive copy strategy to speed up the decoding. Chen et al. (2020) suggest to decoding spans needed to edit only for acceleration. Panthaplackel et al. (2021) present a novel dynamic programming algorithm to allow for span copy during generation. However, this does not break the inherent flaws. In this work, we instead propose a purely non-autoregressive approach for text editing, demonstrating great efficiency benefits over autoregressive counterparts.

**Flexible text generation** To promote more efficient text editing, Malmi et al. (2019); Awasthi et al. (2019) tackle the task as a sequence tagging problem, predicting edit operations with a non-autoregressive decoder. Further developments have been made by GECToR (Omelianchuk et al., 2020), which enhances the method by designing many n-gram and language-dependent edit transformations, e.g., HYPHEN for combining separated tokens, and CAPITAL for capitalizing the first letter of the word, etc. However, the outputs of GECToR are arguably less flexible as its searching space is restricted in a fix small-sized vocabulary. Further, it can hardly produce complex corrections with many insertions, reordering or rephrasing. For this reason, GECToR heavily relies on several (typically 5) rounds of iterative refinements to achieve good performance. FELIX and EDIT5 (Mallinson et al., 2020, 2022) present to use pointer networks and a mask infilling decoder to aid this. But this inevitably incurs more model parameters and more decoding phases.

Compared with the above works, our proposed method is capable of generating very flexible outputs while enjoying similar inference efficiency to GECToR (see § 5.2), which we attribute to the minimal edit operation set in CTC as well as latent edit alignments. We hope our copy-aware CTC will serve as a strong baseline for text editing to elicit further explorations. One possible direction is to improve it with stronger inter-token dependencies, e.g., tree structures (Gui et al., 2023), to further reduce conditional independence (Huang et al., 2022b,a). we leave this to our future work.

# 7 Conclusion

In this work, we propose a novel CTC-based method for non-autoregressive text editing. The key idea is to subsume the editing process with latent CTC alignments. We further make a crucial extension by introducing the copy operation, enabling very effective edits with three main operations only: ADD, KEEP and DELETE. We conduct experiments on GEC and sentence fusion, showing that our method can reach or surpass the current state-of-the-art works while being $4\times$ faster, and generalize well across German and Russian. Moreover, compared with the best performing Seq2Edit model, our method also exhibits very good generation flexibility, which could shed light on further studies on non-autoregressive text editing.

## Limitations

**Decoding burdens** We have noticed that the decoding demands of our method are more substantial than GECToR when limited to a single round of iteration. This can be attributed to the upsampling step before decoding, which must expand the inputs to $T\times$ the original length. Such burdens hinder our capacity to employ more precise yet time-intensive decoding strategies like prefix beam search (Maas et al., 2014). We are going to pursue a more efficient method that requires a relatively small upsampling ratio.

**Precision-Recall tradeoff** One potential limitation of our proposed copy-aware CTC is that it exhibits more pronounced *over-correction* behaviors when compared to GECToR and BART. This is reflected in the fact that our method tends to present higher recalls but relatively lower precisions, which may result in inferior results on some tasks and datasets with a greater emphasis on precisions. We plan to explore strategies that result in better P/R tradeoff in the future.

## Acknowledgements

We would like to thank the anonymous reviewers for their valuable comments, and Prof. Zhenghua Li for very helpful discussions. This work was supported by the National Natural Science Foundation of China (No.62076173), the High-level Entrepreneurship and Innovation Plan of Jiangsu Province (No.JSSCRC2021524), and the Project Funded by the Priority Academic Program Development of Jiangsu Higher Education Institutions.

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

|  | #Sents | Error (%) | Usage |
|---|---|---|---|
| cLang-8 | 2,372,119 | 57.8 | Stage I |
| FCE | 34,490 | 62.6 | Stage II |
| NUCLE | 57,151 | 38.2 | Stage II |
| W&I+LOCNESS | 34,308 | 66.3 | Stage II & III |
| BEA19 | 4,384 | 65.2 | Validation |
| CoNLL14 | 1,312 | 72.3 | Test |

Table 8: Statistics of English GEC data. #Sents and Error (%) refers to the number of sentences and the proportion of erroneous sentences, respectively.

## A Training Details

Table 8 gives the detailed statistics of English GEC data used for training/validating/testing the model.

We train our models for at most 64 epochs based on `roberta-large`. The training process is terminated once the performance on Dev data does not improve after 10 epochs. The experiments are conducted on 1 single Tesla A100 GPU, and it took about 30 hours to finish the Stage I GEC pretraining on cLang-8 data. We train the models with roughly 100,000 tokens per mini-batch and use AdamW for model optimization with $\beta_1 = 0.9, \beta_2 = 0.9, \epsilon = 10^{-12}$. The learning rate is set to $5 \times 10^{-5}$, $5 \times 10^{-6}$ and $1 \times 10^{-6}$ for stage I, II & III, respectively. We also multiply the decoder learning rate by a factor of 10, and warmup the model by 1000 steps for better convergence.

Regarding other tasks and languages, we generally inherit the aforementioned hyperparameters with some adaptations. For multilingual German and Russian, we take `xlm-roberta-large` as the base model. Taking into account that the size of German and Russian data is relatively small, we set the batch size to 10,000 tokens and the learning rate to $2 \times 10^{-5}$ accordingly.