# OpenReview forum: "Non-autoregressive Text Editing with Copy-aware Latent Alignments"
_EMNLP/2023/Conference — EMNLP 2023 Main_

### Official Review · Reviewer_BNoe · 2023-08-05

**Soundness:** 4

**Excitement:**

2: Mediocre: This paper makes marginal contributions (vs non-contemporaneous work), so I would rather not see it in the conference.

**Paper Topic And Main Contributions:**


This paper proposes and evaluates a novel method for efficient non-autoregressive text editing based on adapting connectionist temporal classification (CTC) to incorporate copy actions. Compared to GECToR, the proposed method has better flexibility and language generalization. The main contributions include: (1) Adapting CTC by introducing copy actions like KEEP, ALLOWING direct modeling of edit operations flexibly; (2) Demonstrating strong performance surpassing prior Seq2Edit models on standard GEC and sentence fusion datasets; (3) Exhibiting cross-lingual generalization capabilities by achieving comparable performance to Seq2Seq on German and Russian GEC; (4) Revealing through analysis the robustness of the proposed model across various scenarios and its ability to generate more flexible outputs.

**Reasons To Accept:**

1. It exhibits good cross-lingual generalization capabilities by performing competitively to Seq2Seq models on German and Russian GEC with a single model.

2. The paper demonstrates robust experimental results on GEC and sentence fusion tasks, outperforming previous non-autoregressive methods.

**Reasons To Reject:**

Although this work claims to be the first to adapt CTC for text editing, it is worth noting that other recent works have also explored related non-autoregressive text generation concepts. In my opinion, this work appears to amalgamate pre-existing techniques rather than introducing significant technical novelty.

**Reproducibility:**

4: Could mostly reproduce the results, but there may be some variation because of sample variance or minor variations in their interpretation of the protocol or method.

**Reviewer Confidence:**

3: Pretty sure, but there's a chance I missed something. Although I have a good feel for this area in general, I did not carefully check the paper's details, e.g., the math, experimental design, or novelty.

---

> ### Author Rebuttal · Authors · 2023-08-28
>
> Dear Reviewer BNoe,
>
> Thank you for your insightful feedback.
>
> > Although this work claims to be the first to adapt CTC for text editing, it is worth noting that other recent works have also explored related non-autoregressive text generation concepts. In my opinion, this work appears to amalgamate pre-existing techniques rather than introducing significant technical novelty.
>
> We appreciate your consideration of related concepts in non-autoregressive text generation.
> We want to emphasize that while there have been some explorations in non-autoregreesive machine translation (NAT), our work focuses specifically on adapting CTC for text editing, which presents unique challenges in compared to NAT due to the inherent overlap between inputs and outputs in such tasks.
> Typicial Seq2Edit methods like GECToR address this by employing a bunch of elaborate tags to make cautious edits.
> In contrast, vanilla CTC, unlike both GECToR and our proposed method, lacks an effective mechanism to effectively tackle these challenges, resulting in suboptimal performance.
>
> One of the primary contributions of our work is the incorporation of a copy mechanism into the CTC framework to address the complexities of text-editing tasks, which involves modifying the learning objective to effectively handle the alignment of copy-aware paths.
> We achieve this by providing a novel dynamic programming algorithm, which marginalizes all possible copy-aware alignment paths during training.
>
> We are pleased to direct your attention to Fig. 2 and Table 5, where we provide a visual representation and quantitative analysis of our proposed method's effectiveness.
> We show that our approach not only yields enhanced convergence behavior but also achieves an overall improvement of 1.9 F0.5 scores when compared to vanilla CTC on BEA19 GEC data.

---

### Official Review · Reviewer_M58B · 2023-08-07

**Soundness:** 3

**Excitement:**

4: Strong: This paper deepens the understanding of some phenomenon or lowers the barriers to an existing research direction.

**Paper Topic And Main Contributions:**

The authors present a non-autoregressive (NAR) text editing algorithm which extends CTC with a copy operation, and models editing with latent alignments from CTC. NAR output with flexible length is enable by 4x upsampling input length. The proposed approach extends upon recent work in the seq2edit paradigm, where a target sequence is generated from a source sequence by applying a series of edit operations. Latent edit operations are marginalized over during training to maximize target sequence likelihood.

**Reasons To Accept:**

- Novel method of using CTC to model GEC with NAR decoding
- Clear exposition and relatively thorough evaluation


**Reasons To Reject:**

- Insufficient exposition of the Gamma function implementation, at least a few sentences should be allocated to explain how this is implemented
- lower precision than GECTOR (but somewhat better recall results in overall better F1)
- large expansion of the input length makes decoding slower

**Reproducibility:**

4: Could mostly reproduce the results, but there may be some variation because of sample variance or minor variations in their interpretation of the protocol or method.

**Reviewer Confidence:**

4: Quite sure. I tried to check the important points carefully. It's unlikely, though conceivable, that I missed something that should affect my ratings.

**Typos Grammar Style And Presentation Improvements:**

this paper has many typos, needs careful proofreading
a few of them:
030-034 - Table 1 does not show text summarization (it shows GEC and sentence fusion)
040 fasion → fashion
230 log-likelyhood → log-likelihood
251 predicts → predict
252 parallely → in parallel

---

> ### Author Rebuttal · Authors · 2023-08-28
>
> Dear Reviewer M58B,
>
> Thank you for your very constructive feedback.
> We will try our best to improve the writing and polish the paper according to your comments in the next version.
>
> > Insufficient exposition of the Gamma function implementation, at least a few sentences should be allocated to explain how this is implemented
>
> Thank you for highlighting this aspect.
> In Table 2, we give a toy example for the sentence ***I like an dog***, wherein ***K K K K $\emptyset$ $\emptyset$ $\emptyset$ dogs*** represents one of the possible sequences returned by the Gamma function.
> To enhance clarity, we recognize the need to offer a more comprehensive explanation of the implementation.
> We will add more examples to mitigate any potential ambiguities in the upcoming version.
>
> > lower precision than GECToR (but somewhat better recall results in overall better F1), large expansion of the input length makes decoding slower
>
> Regarding your observation about precision compared to GECToR, we acknowledge this trade-off and would like to emphasize that our model's strength lies in its superior F0.5 scores across various tasks and languages.
> This underlines its effectiveness in a broader range of contexts.
> We also wish to draw special attention to the inherent advantage of our model in generating more flexible outputs when compared to GECToR (see Section 5.2).
>
> Furthermore, we'd like to clarify that despite the challenges posed by the upsampling layer, our proposed model maintains a commendable speed even when compared to GECToR.
> Notably, GECToR also operates in a non-autoregressive manner, and our method's efficiency stems from the significantly reduced number of refinement iterations (2 v.s. GECToR's 5).
> We acknowledge the importance of efficient decoding methods and recognize that future research could delve into strategies that accommodate inputs of shorter lengths while preserving performance.

---

### Official Review · Reviewer_L81y · 2023-08-11

**Soundness:** 4

**Excitement:**

4: Strong: This paper deepens the understanding of some phenomenon or lowers the barriers to an existing research direction.

**Paper Topic And Main Contributions:**

This paper presents a novel approach to text editing. The proposed method is based on a Seq2Edit approach, which relies on applying a sequence of edit operations over the input, as opposed to the previous generation of methods based on a classical Seq2Seq approach. The advantage of the Seq2Edit approach is that it is much faster than Seq2Seq. The main contribution of this work is applying CTC (connectionist temporal classification) to this task. CTC is in general used in ASR. The method proposed here interprets the CTC alignments as a series of edit operations that can then be executed over the input sequence. Crucially, the authors introduce a “KEEP” edit  operation (in addition to the traditional “ADD” and “DELETE”), which denotes that an input token can be directly copied to the output. The authors credit this modification with the important speed gains of their method. Special attention is given to output flexibility and generalizability to other languages.

The method is evaluated on two text edit tasks: grammatical error correction and sentence fusion. It is evaluated against a wide range of existing methods. In both cases, the proposed method is on par or better than the reported existing methods. Additionally, on the grammatical error correction task, the proposed method is 30x faster than the Seq2Seq model at inference time, and there is also evidence that the modified CTC approach converges faster at training time than the standard CTC approach. The authors also investigate the relationship between model performance and WER, as well as the output flexibility. They test the multilingual generalization capacities of the method by applying it to German and Russian with encouraging results.

-----------------------------------------

__*Post-rebuttal comment*__
The authors have addressed my concerns in a thorough and thoughtful manner. Their answers strengthen my impression that this is a solid and well thought out contribution and will make a good paper once the suggestions are integrated. I am keeping my grades unchanged because my concerns were minor and the grades were based on my perception of the scope and the contribution of the paper rather than the concerns themselves.

**Questions For The Authors:**

A. line 072: A clarification question: the criticism of the existing Seq2Edit models states that these models rely on linguistic features to customize the edit actions. It remains unclear for me if this is meant in a general way (needing to establish the pattern of house : houses, chair : chairs as meaningful) or if the method explicitly relies on POS tags. Could you clarify this?

B. Line 204: How was the value of T=4 decided?

C. Line 412: “the predicted alignments of our proposed CTC variant are more in agreement with human opinions than those of vanilla CTC.”  This is a strong statement. What is it based on?

D. Table 6: when comparing your method to pre-existing models, are you rerunning the evaluation for the pre-existing models or reporting the results from pre-existing papers? If you are rerunning the evaluations, why are there missing P and R scores for gT5 and mT5? If you are reporting results that were published before, it should be stated clearly and the papers correctly cited. This stands for all evaluations.

E. Line 470: I found the section “Regarding flexibility” very hard to follow. Most comments seem unnecessarily convoluted. E.g., “We observe that GECToR performs awkward in OOV cases, but for the first two types, GECToR has very good coverage with only 3.7% gold VERB references not covered. GECToR performs favorably better than BART and CTC in this situation.”. Does this mean the following: “GECToR has extremely bad performance on OOV VERB edits (0 F0.5 score). However, there is a very low proporition of OOV VERB edits in the reference corpus. Therefore, GECToR’s performance on VERB edits is not overly affected by this shortcoming.”? Could you clarify the rest of the comments? I am interested in your results here but am not sure I am understanding the text correctly.

**Reasons To Accept:**

**NB**

I would like to specify that I was recruited as an emergency reviewer and that the topic of the paper is not in my domain of expertise. Therefore I have a hard time evaluating the novelty and the impact of the method beyond what is presented in the paper. This will be reflected in my reviewer confidence score.

----------------------------------------

In general, this seems like a well thought out work. The claims are supported by a wide range of experiments, and the paper provides a lot of information on how it was all done. In particular:

1. This paper presents an approach that is novel and effective, leading to interesting improvements both in terms of results and computational performance. An additional positive aspect is that the method is based on adapting an approach from a different NLP domain (ASR). The adaptation is straightforward and clearly described.
2. Solid evaluation of the proposed method. The method is evaluated on two different tasks, against multiple existing models. Several targeted evaluations are done in order to examine different aspects of model performance.
3. Generalizability to other languages. The authors state explicitly that their goal is to create a method that is language-independent and can be applied to other languages. They test this by applying their model to Russian and German with satisfying results.
4. The authors indicate that their code is available on github and say that the model will be released.

**Reasons To Reject:**

I have not identified any major issues that would warrant rejection. However, as stated above, this is not my domain of expertise. I do have a number of smaller issues to raise, which will be done in the questions and typo/style suggestions.

**Reproducibility:**

3: Could reproduce the results with some difficulty. The settings of parameters are underspecified or subjectively determined; the training/evaluation data are not widely available.

**Reviewer Confidence:**

1: Not my area, or paper was hard for me to understand. My evaluation is just an educated guess.

**Typos Grammar Style And Presentation Improvements:**

In general, the paper is very dense and occasionally hard to follow. While I salute the richness of the information provided, the paper may benefit from a more careful selection of the content. It would make it easier to read. In any case, I strongly suggest a major editing pass focused on making sure that the main ideas and concepts are clearly presented and easy to grasp.

Abstract: CTC, GEC – consider spelling these out in the abstract. The first occurrences in the paper body are too far to be useful.

Section 5: it would be useful to state explictly on which task the comparisons in Section 5.1 are done. This can be deduced from the table and figure captions, but it makes the reading more difficult.

The dataset sizes are not given in the paper body. For the GEC task, they are given in the appendix,  but they are not given at all for the sentence fusion task. I would suggest making this information available in the paper itself: readers who are not familiar with the datasets would appreciate it.

Related Work: Consider moving this section after the introduction. If the introduction and the related work were closer together, it would allow you to shorten both of them, since some of the information is present in both.

Consider expanding all contractions such as “it’s” (e.g. line 103)

Table1: in the GEC example, in the target sentence, should the first ‘to’ also be in blue?

061: meanwhile orders of magnitude faster → while being orders of magnitude faster.

064: there arise several challenges → consider replacing by ‘several challenges arise’

112: four-folds → four-fold

115: to our best knowledge → to the best of our knowledge

120: favorably better → better

149: factorize → factorizes

164: from the regressive counterpart → from its regressive counterpart

Figure 1 is not referenced in the text of the paper.

Line 171: Consider clarifying why in ASR CTC assumes that the input is longer than the output;, and why that is not the case in text generation.

173: resulting → resulting in

180: grants the power of CTC → grants power to CTC?

190: give → given

196: once getting the hidden states → once the hidden states are obtained

216: that → of

221: explicit model → an explicit model

222: We in this work propose → In this work, we propose

230: likelyhood → likelihood

252: parallely → parallelly

253: Table. 2 → Table 2

277: representive → representative?

307: which has huge impact on final results → which has a huge impact on the final results

310: for better access of the results → for more comparable results

319: our models → our model

315: is superior to BART on all datasets by 0.4, 0.1, and 2.2 scores → remove ‘scores’. Same at line 326

345: that → than

363: 1 EM score → 1 EM point

384: precisions → precision

399: while → with

401: with about 1.5 final F0.5 gains → with a final gain of about 1.5 F0.5 points

403: Alighnment → alignment

434, 436: scores → points

464: resulting a inferior → resulting in an inferior

472: are → is

Table 7 needs a clearer caption. The explanation in the text is hard to follow too.

506: related works → related work

566: we leave this as our future works → We leave this to our future work.

577: current state of the art works with over 30x times faster → current state of the art while being 30x faster

582: shed lights → shed light

---

> ### Author Rebuttal · Authors · 2023-08-28
>
> Dear Reviewer L81y,
>
> Thank you for your detailed and helpful comments.
> We are committed to improving the quality of our paper based on your suggestions.
> Addressing the typos and enhancing the clarity of the writing are key aspects that we will focus on in the next version.
>
> > It remains unclear for me if this is meant in a general way (needing to establish the pattern of house : houses, chair : chairs as meaningful) or if the method explicitly relies on POS tags. Could you clarify this?
>
> We would like to clarify that the representative Seq2Edit-based GECToR method encompasses both scenarios you mentioned.
> The tagger introduces `SINGULAR`/`PLURAL` tag to handle the singular and plural conversion of nouns, `LOWER`/`UPPER` to handle the case conversions, etc.
> In more intricate cases, GECToR leverages the power of POS tags to make nuanced distinctions.
> For instance, it employs the `VB-VBZ` tag transformation to convert verbs like "make" to "makes", and similarly employs `VBZ-VBG` to transform words like "thinks" to "thinking".
>
> > How was the value of T=4 decided?
>
> In our preliminary experiments, we have tried different values of `T`, ranging from 2 to 8, as detailed in Table 5.
> We identified that setting `T=4` yields the best performance on BEA19 data without sacrificing too much speed.
> So we directly fix the value of `T` to 4 in the latter experiments.
> We will clarify this in the next version. Thank you!
>
> > “the predicted alignments of our proposed CTC variant are more in agreement with human opinions than those of vanilla CTC.” This is a strong statement. What is it based on?
>
> Considering the examples in Fig. 3, we can see that when empowered by the copy mechanism, our CTC variant demonstrates the capacity to properly generate the alignment for phrases like "couldn't" and "couldn't she".
> This achievement mirrors the human decision-making process for alignment, which entails identifying common tokens between source and target sentences, and using them as reference points for alignment.
> We posit that the direct copying capability in our model reduces over-correction tendencies, ultimately leading to enhanced results.
> We will enhance the clarity of our explanations in the subsequent edition. Thank you for raising this concern.
>
> > when comparing your method to pre-existing models, are you rerunning the evaluation for the pre-existing models or reporting the results from pre-existing papers? If you are rerunning the evaluations, why are there missing P and R scores for gT5 and mT5? If you are reporting results that were published before, it should be stated clearly and the papers correctly cited. This stands for all evaluations.
>
> For our comparison with pre-existing models, we have sourced the results directly from the original papers without re-running the evaluations.
> In Table 7, we have referenced the papers for mBART, gT5, and mT5 in a succinct manner due to space constraints.
> We are committed to making these citations more prominent and clearly explaining our approach in the upcoming version. Thank you for bringing this to our attention.
>
> > I found the section “Regarding flexibility” very hard to follow. Most comments seem unnecessarily convoluted. E.g., “We observe that GECToR performs awkward in OOV cases, but for the first two types, GECToR has very good coverage with only 3.7% gold VERB references not covered. GECToR performs favorably better than BART and CTC in this situation.”. Does this mean the following: “GECToR has extremely bad performance on OOV VERB edits (0 F0.5 score). However, there is a very low proporition of OOV VERB edits in the reference corpus. Therefore, GECToR’s performance on VERB edits is not overly affected by this shortcoming.”? Could you clarify the rest of the comments? I am interested in your results here but am not sure I am understanding the text correctly.
>
> Thank you for your kind clarification.
> In Table 7, the OOV column presents a comparison among the three methods based on the prediction scores for instances where token-level GECToR edits are unable to be applied.
> It is noteworthy that, despite the challenges, GECToR does have the potential to generate OOV predictions through the combination of multiple edits, as evidenced by the example in Footnote 4.
> One crucial aspect we wish to underscore is that GECToR is severely restricted by the limited vocabulary space compared with CTC and BART.
> Consequently, the overall performance of GECToR is strongly influenced by the proportion of OOV cases.
> Notably, the most prominent prevalence of OOVs emerges in the `OTHER` category (8\% of gold references), causing GECToR to exhibit its weakest overall performance (56.1) in this specific category.
> In contrast, both CTC and BART exhibit a higher capacity to generate adaptable outputs, thereby yielding higher F0.5 scores.
> We will revise our writing to make it more clear.
>
> > Consider clarifying why in ASR CTC assumes that the input is longer than the output, and why that is not the case in text generation.
>
> In ASR, the input consists of a sequence of acoustic units, naturally exceeding the length of the output token sequence.
> On the contrary, in text generation tasks such as English-to-German translation, both input and output are sequences of tokens, and the former is not inherently longer than the latter.
> To fulfill the assumptions underlying the CTC methodology, we indeed employ an upsampling technique to magnify the input sequence.
> This aligns with the requirement that the input surpasses the output in terms of length.
> We promise to provide a clearer and more detailed explanation in the next version, thank you!

---

### Meta-Review · Area_Chair_SG8c · 2023-09-24

**Recommendation:** 4

**Metareview:**

The paper presents a non-autoregressive seq2edit approach for text editing in order to improve inference speed. The novel approach is based on connectionist temporal classification, traditionally used in ASR. The method is evaluated in two tasks: grammatical error correction and sentence fusion.

Reviewers agree that the method is effective, clearly described and straightforward to apply. There is also a solid evaluation procedure, comparing the approach against multiple existing models. Evaluation is also cardied out in other two languages other than English with satisfying results. Finally, the paper also includes a through analysis of the robustness of the proposed model across various scenarios and its ability to generate more flexible outputs.

A main concern is related to the novelty of the approach, since similar ones have been proposed for machine translation. The authors have clarified how their apporach differns, and are encouraged to highlight these differences in the paper. Some concerns were raised regarding details of certain model components, which were clarified during rebuttal. There is also a concern that the paper is dense, and some suggestions were provided for improving the clarity. Authors are encouraged to include these recommendations and clarifications in the final version.

---

### Decision · Program_Chairs · 2023-10-07

**Decision:**

Accept-Main

**Comment:**

The paper presents a non-autoregressive seq2edit approach for text editing in order to improve inference speed. The novel approach is based on connectionist temporal classification, traditionally used in ASR. The method is evaluated in two tasks: grammatical error correction and sentence fusion.

Reviewers agree that the method is effective, clearly described and straightforward to apply. There is also a solid evaluation procedure, comparing the approach against multiple existing models. Evaluation is also cardied out in other two languages other than English with satisfying results. Finally, the paper also includes a through analysis of the robustness of the proposed model across various scenarios and its ability to generate more flexible outputs.

A main concern is related to the novelty of the approach, since similar ones have been proposed for machine translation. The authors have clarified how their apporach differns, and are encouraged to highlight these differences in the paper. Some concerns were raised regarding details of certain model components, which were clarified during rebuttal. There is also a concern that the paper is dense, and some suggestions were provided for improving the clarity. Authors are encouraged to include these recommendations and clarifications in the final version.